# Prevention or Amelioration of Autism-Like Symptoms in Animal Models: Will it Bring Us Closer to Treating Human ASD?

**DOI:** 10.3390/ijms20051074

**Published:** 2019-03-01

**Authors:** Asher Ornoy, Liza Weinstein-Fudim, Zivanit Ergaz

**Affiliations:** 1Laboratory of Teratology, Department of Medical Neurobiology, Hebrew University Hadassah Medical School, Jerusalem 9112001, Israel; liza.weinstein-f@mail.huji.ac.il; 2Neonatology Department, Hadassah Hebrew University Medical Center, Jerusalem 9112001, Israel; Zivanit@hadassah.org.il

**Keywords:** animal models of ASD, rodents, amelioration, prevention, VPA, oxytocin, SAM, PUFAs, human ASD

## Abstract

Since the first animal model of valproic acid (VPA) induced autistic-like behavior, many genetic and non-genetic experimental animal models for Autism Spectrum Disorder (ASD) have been described. The more common non-genetic animal models induce ASD in rats and mice by infection/inflammation or the prenatal or early postnatal administration of VPA. Through the establishment of these models, attempts have been made to ameliorate or even prevent ASD-like symptoms. Some of the genetic models have been successfully treated by genetic manipulations or the manipulation of neurotransmission. Different antioxidants have been used (i.e., astaxanthin, green tea, piperine) to reduce brain oxidative stress in VPA-induced ASD models. Agents affecting brain neurotransmitters (donepezil, agmatine, agomelatine, memantine, oxytocin) also successfully reduced ASD-like symptoms. However, complete prevention of the development of symptoms was achieved only rarely. In our recent study, we treated mouse offspring exposed on postnatal day four to VPA with *S*-adenosine methionine (SAM) for three days, and prevented ASD-like behavior, brain oxidative stress, and the changes in gene expression induced by VPA. In this review, we describe, in addition to our data, the existing literature on the prevention/amelioration of ASD-like symptoms. We also discuss the possible mechanisms underlying some of these phenomena. Finally, we describe some of the clinical trials in children with ASD that were carried out as a result of data from animal studies, especially those with polyunsaturated fatty acids (PUFAs).

## 1. Introduction

Autism spectrum disorder (ASD) is a heterogeneous group of neurobehavioral deviations that occur in about 1% of children. More recent surveys in the United States (US) have reported a prevalence of 1/68 or even 1/40, with a higher rate in boys than in girls [1,2,3]. Genetic and environmental factors seem to be responsible, but the exact etiology and pathogenesis are not well defined [4]. Treatment is “symptomatic”, as pharmacological treatment alleviates the different symptoms (i.e., treatment of the attention deficit hyperactivity disorder (ADHD)-like behavior by stimulants of the mood disorder by antipsychotic drugs) and behavioral–educational treatment is aimed to improve the communication abilities.

In many defined human syndromes, autistic-like behavior may be a major clinical symptom. For example, Fragile X syndrome with the *FMR1* gene residing in the X chromosome and preventing the synthesis of the RNA binding protein fragile X mental retardation protein (*FMRP*), or Rett syndrome, which affects girls where there is a mutation of the *MECP2* gene, thus interfering with the normal methylation of cytosine phosphate guanine (CpG) islands and the activation/deactivation of gene expression. There are other, less common genetic diseases with autistic-like traits. Since there are several animal models for these different diseases, experimental animals, especially mice and rats, are being used for the study of ASD, enabling studies on the etiology, pathogenesis, and possible prevention and treatment modalities of ASD.

## 2. Animal Models for Autism Spectrum Disorder (ASD)

### 2.1. Genetic Animal Models

There are two types of animal models for ASD. First, there are those induced by genetic manipulations, especially in mice, representing well-defined human diseases. One example is knockout mice that lack the *FMR1* gene, and thus mimic human Fragile X. Neurodevelopmental studies have shown autistic-like behavior in these animals [5]. A second example is of mice with mutations in the gene encoding for *MECP2* that develop typical Rett syndrome and autistic-like behavioral features [6]. Another relatively common genetic model is the *BTBR T + tf/J* (*BTBR*) mouse model where the animals show impaired social interaction, impaired communication, and impaired and repetitive motor movements [7]. These and many other animal models also have the advantage that they enable carrying out molecular and pathological studies on the specific brain changes, and thus serve as a tool for the comprehensive study of the exact changes underlying these abnormal behaviors. Moreover, they also enable to carry out genetic manipulations to try and reverse these ASD-like behavioral changes [5,6,7,8,9,10,11].

### 2.2. Correction (Reversal) or Amelioration of ASD-Like Behavior in Genetic Models of ASD in Mice and Rats

There are many examples of correction in genetic models of ASD-like behavior. The more important ones will be discussed briefly.

Examples of Correction (reversal):

#### 2.2.1. Correction of the Mouse Model of Rett Syndrome

Guy et al. [8] postulated that in mice with Rett syndrome, the neurons do not die in spite of the *MECP2* mutation, and that manipulation with the *MECP2* gene might reverse the situation. Indeed, they triggered the re-expression of the *MECP2* gene in a mouse model of *MECP2* deficiency, and reversed the autistic-like behavioral changes as well as the abnormal neurological symptoms that are typical of Rett syndrome.

#### 2.2.2. Correction in the Mouse Model of Fragile X with Autistic-Like Behavior

Correction of the neurological and behavioral impairment in mice was achieved by either removal of the gene encoding for p70 S6 kinase 1 [5], early social enrichment via enhanced maternal care of the *FMR1-Ko* mouse offspring [9], or treatment with polyunsaturated fatty acid (PUFAs) [10] which, without treatment, exhibit fragile X-like behavioral phenotypes. For example, Pietropaolo et al. [10] studied the possible effects of omega-3 fatty acids (n-3 PUFA) on the ASD-like behavior in the *FMR1-Ko* mouse, which is a model of fragile X syndrome. They found that daily supplementation of n-3 PUFAs improved the social interaction, emotionality, and non-spatial memory of the fragile X mice and normalized (reduced) some of the neuroinflammatory changes in the brain.

#### 2.2.3. Oxytocin Reversal/Amelioration

Oxytocin is a neuropeptide that is known to have an important role regarding maternal behavior toward progeny, including attachment and mother–infant bonding. Studies have shown that oxytocin deficiency may cause social behavioral deficits [11,12,13]. Teng et al. [14] studied the effects of oxytocin in two strains of mice with ASD-like behavior: the *C58/J* and the *Grin 1* knockout mouse, which were both serving as models for ASD. These studies followed previous studies of this group and of others who demonstrated the beneficial effects of the chronic administration of oxytocin in reducing the social deficits of several genetically manipulated strains of mice with ASD-like behavior. Four injections of oxytocin administered every second day improved the social behavior of the mice with ASD-like behavior at adolescence and at adulthood [14]. These reductions of ASD-like symptoms lasted for several weeks. Therefore, the authors considered the effects of oxytocin as “reversal”. In a later study, Teng et al. [15] have found that in *C58/J* mice with autistic-like social behavior and repetitive stereotypy, oxytocin administration caused significant social improvement at adolescence. If administered at adulthood, the beneficial effects of oxytocin lasted even two weeks following the cessation of treatment.

### 2.3. Studies on BTBR Mice

#### 2.3.1. Characteristics of this Strain

This genetic strain of mice with autism-like behavior is commonly used for the study of ASD. The animals are characterized by social behavioral impairment as well as immune system dysfunction, and display several single nucleotide polymorphisms [7]. They have both social communication impairment and repetitive movements. They also have an indifference to sweets. They exhibit several changes in genes related to brain functions, including brain-derived neurotrophic factor (*BDNF*) in the hippocampus and cerebral cortex, and changes in several synaptic proteins. They also demonstrate changes in DNA methylation (epigenetic changes) in the cerebellum. This mouse model of ASD was used by several investigators for the development of treatment modalities.

#### 2.3.2. Treatment by Insulin-Like Growth Factor 2 (*IGF-II*) and Beta Carotene

Steinmetz et al. [16] used male *BTBR* mice to evaluate the possible effects of insulin-like growth factor 2 (*IGF-II*) on the reduction of ASD-like behavioral changes that were manifested by cognitive impairment, social behavioral deficits, and repetitive behaviors. *IGF-II* is a polypeptide that crosses the brain–blood barrier and is used as a cognitive enhancer. When these mice were injected with IGF-II prior to the behavioral testing (B6 mice were used as controls), reversal of the typical abnormal behaviors was achieved. Memory deficits in the *BTBR* mice were also reversed by *IGF-II*.

Avraham et al. [17] administered orally 0.1 mg/kg/day, 1.0 mg/kg/day, or 5.0 mg/kg/day of beta carotene during postnatal days one through seven to newborn *BTBR* or *BALB/c* mice that exhibited autistic-like behavior. Various behavioral tests were carried out at the age of two to three months. The administration of beta carotene increased social interaction and communication of the mice, and normalized some of the other behaviors of the mice of both strains.

#### 2.3.3. Effects of Modulation of Neurotransmission

Silverman et al. [18] used *BTBR* mice with ASD-like symptoms to try and ameliorate the ASD-like symptoms by manipulating excitatory neurotransmission. They injected *BTBR* mice with the glutamate modulator GRN-529, which is a selective modulator of the metabotropic glutamate receptor subtype 5, and were able to partially reverse the abnormalities of social behavior and alleviate the repetitive movements (i.e., reduce the stereotyped vertical jumping). These ameliorations of the symptoms were achieved by a non-sedating dose of three mg/kg body weight of GRN-529 with only about 50% occupancy of the glutamate receptor.

Since one of the hypotheses for the clinical characteristics of ASD is an imbalance between excitation (mainly represented by glutamate) and inhibition (mainly represented by gamma-aminobutyric acid (GABA)) in the brain, Yoshimura et al. [19] investigated the possible effects of modulating nicotinic and *GABA_A_* receptor subtypes on ASD-like behavior in *BTBR* mice. They found that the pretreatment of *BTBR* mice 30 min before testing with several compounds that have selective activities at the *GABA_A_* receptor and α7 nicotinic acetylcholine receptor (all were developed in their laboratory) improved some of the behavioral deficits of the *BTBR* mice. The compound that had selective activity at the *GABA_A_* receptor improved the social behavior, but not the self-grooming. On the other hand, the α7 nicotinic acetylcholine receptor positively affected both self-grooming and social behavior. None of these compounds affected locomotor activity. The effects were dose-dependent.

#### 2.3.4. Effects of Bone Marrow Transplantation in *BTBR* Mice

As described above, there are many signs of increased neuroinflammation in the *BTBR* mouse, which is considered one of the factors responsible for the ASD-like behavior in these animals. Indeed, Schwarzer et al. [20] irradiated *BTBR* mice to achieve bone marrow ablation, and then injected them with bone marrow cells from normal *C57BL* mice. They found that bone marrow from normal *C57BL* mice improved the sociability of the *BTBR* mice. They further repeated the studied on *C57BL* mice transplanted with bone marrow cells from *BTBR* mice, and observed the deterioration of the social behavior of these mice, which was manifested by increased repetitive grooming activity, thus demonstrating the importance of the immune system in social behavior.

### 2.4. Non-Genetic Animal Models of ASD-Like Behavior

The second type of ASD-like animal models is environmentally-induced ASD-like behavior. In these different models, attempts to affect the offspring were carried out either by manipulating the pregnant dams or directly treating the offspring early after birth. Brain development during early postnatal days in mice and rats is equivalent to the developmental stages of the human fetal brain in the third trimester of pregnancy [21]. Both types of procedures may result in autistic-like behavioral changes. These models either use specific chemicals (i.e., valproic acid) or infection/inflammation to promote the autistic-like behavioral features; both of these models will be discussed in detail, as the issue of prevention, alleviation, and treatment without any genetic manipulations can be studied only in such models.

## 3. Animal Models of Infection/Inflammation

### 3.1. Induction of ASD-Like Behavior

An inflammatory/infectious etiology of ASD was suspected in humans, because large population-based studies found an increased rate of ASD in the offspring of women with fever and infection in pregnancy, which was apparently related to maternal inflammatory processes. Hence, maternal immune activation seems to play a role in the etiology of neurodevelopmental problems [22].

Therefore, animal models are used to associate maternal infection/inflammation during pregnancy and abnormal histological and behavioral findings in the offspring, and then correlate them with behavioral impairment. Studies have shown [23] that maternal immune activation induced changes in several interleukins in the fetal brain, as well as morphological changes in the hippocampus and cerebral cortex. Inflammation was generally triggered by the injection of immunogens during pregnancy, such as lipopolysaccharides (LPS) [24] and polyinosinic–polycyclidilic acid (Poly IC, a double-stranded RNA that mimics viral infection) [25]. These models enabled the study of the pathological and behavioral changes and the molecular and biochemical mechanisms involved in the nervous system pathology, especially in the hippocampus and cerebral cortex. Most of these animal models evaluated the inflammatory process, rather than specific viruses.

Xuan et al. [26] found in a murine model that male and female offspring from mothers injected with LPS (endotoxin) or Poly IC during mid-gestation displayed autistic-like behaviors at adulthood. They also found differences in the results of behavioral tests between genders, in which males showed more repetitive behaviors than females.

The influenza virus murine model and Borna disease virus in mice and rats are examples of inflammation-induced ASD-like behavior. Influenza virus in pregnant mice at mid-gestation resulted in behavioral abnormalities in the offspring, demonstrating deficits in social interaction and prepulse inhibition [27]. Neonatal Borna virus infection in rats resulted in decreased play activity and at adulthood, a stereotypic behavior associated with a decrease in exploratory activity [28].

### 3.2. Alleviation of ASD-Like Behavior in These Animal Models

Only a few attempts have been reported of the alleviation of the ASD-like behavior in animal models of inflammation/infection-induced ASD. Kirsten et al. [29] demonstrated in rats that the injection of lipopolysaccharide endotoxins (LPS) on day 9.5 of gestation induced autism-like behavior in the offspring. Pioglitazone, which is an agonist of the peroxisome proliferator-activated receptor gamma (*PPAR gamma*) with anti-inflammatory properties, administered to the offspring of endotoxin-treated dams from postnatal days 21 to 29 alleviated the autistic-like behavioral changes, as evidenced from studies on socialization, communication, and ultrasonic vocalization at adulthood. In addition, the elevated plasma interleukin-6 (*IL-6*) levels that were found in the endotoxin-exposed offspring were reduced by the administration of pioglitazone.

Choi et al. [30] found that maternal immune activation in mice affected fetal cerebral cortical development, inducing autistic-like behavior. They found that *IL-17α* was the main fetal cytokine responsible for the abnormalities regarding the cortical development of the fetal brain. Pretreatment of the dams with *IL-17α* blocking antibodies prevented maternal rise in the levels of *IL-17α* and also diminished the rise in fetal *IL-17α*. In addition, this also ameliorated the ASD-like behavioral abnormalities.

Vuillermot et al. [31] administered Poly IC to pregnant mice on day nine of gestation, inducing ASD-like behavior in the offspring, especially repetitive movements, anxiety, and abnormal social behavior. By pretreatment with the active metabolite of vitamin D (1,25(OH)D_3_) all of the ASD-like behavioral changes were prevented.

## 4. Animal Models of Chemically-Induced ASD-Like Behaviors

### 4.1. Valproic Acid (VPA) and ASD in Human and in Rodents

Valproic acid (VPA) is associated with a high risk of ASD. Christianson et al. [32] were apparently the first to describe a possible association between intrauterine exposure to VPA and ASD. Later, Williams et al. [33] described several additional children with the typical facial features of valproate syndrome and autism. Other investigators [34,35,36,37] further reported on the association of VPA use during pregnancy and ASD in the offspring. Rasalam et al. [34] found that ASD, according to Diagnostic and Statistical Manual of Mental Disorders (DSM IV) criteria, was present in five of 56 children prenatally exposed to VPA monotherapy (8.9%), and the rate increased to 11.7% among children exposed prenatally to sodium valproate in combination with other AEDs. Moore et al. [37] found that four out of 57 children prenatally exposed to VPA had autism, and an additional two had Aspergers. Bromley et al. studied the outcome of 632 children prenatally exposed to antiepileptic drugs; of them, 64 were exposed to VPA monotherapy. They found that four of these 64 children (6.3%) had autism, and another child with ASD was exposed to VPA polytherapy. This is seven times higher than in the general population [35]. There was no association with other antiepileptic drugs. Finally, Christensen et al. [36] reviewed the Danish National Population Register using prescription data and found a tripling of the rate of ASD among 508 children prenatally exposed to VPA.

Several investigators developed VPA-induced ASD-like behavior in mice and rats [38,39]. In these models, various anatomical and functional alterations in the cerebral cortex and cerebellum were described, some of them similar to the changes that had been described in the brains of children with ASD. Among these changes were reduced numbers of cerebellar Purkinje cells, damage to cranial nerve nuclei, and synaptic changes in the cerebral cortex [40,41]. Studies have shown that VPA can induce ASD-like behavior in mice and rats at almost any stage of pregnancy, as well as in the first two postnatal weeks.

### 4.2. Studies on the Prevention/Amelioration of ASD-Like Behavior in Animals Treated with VPA 

Many studies have been published in the last several years that have tried to prevent, reverse, or ameliorate ASD-like symptoms by the administration of “protective” agents close to the time of VPA administration or thereafter, trying to the ASD-like symptoms. Several studies were aimed at reducing the brain oxidative stress, and thus alleviating, (generally temporarily, lasting as long as treatment continued), the neurobehavioral symptoms. Other studies aimed to prevent the development of ASD-like behavior. From the results of these studies, it is sometimes difficult to judge whether the behavioral improvement was temporary, thus demonstrating the alleviation of symptoms, or whether it induced the real prevention and reversal of the ASD-like behavior caused by VPA. Hence, they will be dealt together (Table 1).

#### 4.2.1. Amelioration by Antioxidants

Banji et al. [42] treated 14-day-old mice with a single injection of 400 mg/kg of VPA, and induced ASD-like behavior. The daily administration, from the day of VPA treatment, of 75 mg or 300 mg of green tea extract with known anti-oxidative effects ameliorated the ASD-like behavioral changes in the VPA-treated mice, with a more pronounced effect of the higher dose of 300 mg/day. There was also a marked improvement in the histopathological cerebellar changes induced by VPA, as the damage to Purkinje cells induced by VPA administration were not found after the administration of the higher dose. The authors attributed these protective effects to the antioxidant activity of green tea.

Al-Amin et al. [43] injected pregnant mice with 600 mg/kg of VPA on day 12.5 of gestation, and produced ASD-like behaviors in the offspring, as observed on postnatal day 25. When the offspring were treated from postnatal day 26 to day 56 with daily injections of high doses (two mg/kg body weight) of astaxanthin, which is a potent antioxidant, there was an improvement in the ASD-like behaviors and a significant reduction in the degree of oxidative stress.

Pragnya et al. [44] injected 14-day-old mice with 400 mg/kg of VPA and produced autistic-like behavior. In addition, the brain manifested increased oxidative stress and some histopathological changes in the cerebellum. Concomitant treatment of the VPA-injected mice with five mg/kg or 20 mg/kg of piperine (a known antioxidant) during postnatal days 13 to 40 improved the ASD-like behavioral features, and reduced brain oxidative stress by increasing glutathione levels and decreasing the total nitrite levels to those of the control animals. Piperine also increased the number of cerebellar Purkinje cells to almost normal levels. The higher dose of piperine was more effective than the lower. Thus piperine had neuroprotective and preventive activity, which was apparently due to its antioxidant properties.

#### 4.2.2. Protection by Polyunsaturated Fatty Acids (PUFAs)

Yadav et al. [45] injected subcutaneously 14-day-old rats with 400 mg/kg of VPA or with VPA and oral administration of the PUFAs α linoleic acid (ALA) or γ linoleic acid (GLA), and evaluated the effects of these fatty acids on the results of behavioral tests assessing autism-like behavior. The doses of PUFAs were three mL/kg body weight administered daily from day 15 to day 25. They assessed the levels of acetyl cholinesterase, catalase, superoxide dismutase (SOD), protein carbonyl, glutathione and lipid peroxidation in the brain, which are all markers of oxidative stress. Both ALA and GLA decreased the levels of the oxidative stress markers in the VPA-treated rats and alleviated some of the neurobehavioral changes induced by VPA. However, GLA was more effective compared to ALA, because it also protected the cerebellum from the VPA-induced pathological injuries. VPA increased cerebellar neuronal loss and microglial activation, while the addition of GLA decreased VPA-induced neuronal loss to almost control levels. Thus, both GLA and ALA were protective, and GLA was more effective than ALA.

#### 4.2.3. Alleviation by Agents Affecting Brain Neurotransmitters

In a series of studies Kumar et al. [46,47,48] found that the injection of VPA to rats on day 12.5 of gestation induced autistic-like behavior in the offspring. The daily administration of memantine, (an *N*-methyl-d-aspartate (*NMDA*) receptor modulator), minocycline (a modulator of social behavior in rats), or agomelatine (a selective melatonin membrane type 1 and type 2 (*MT1* and *MT2*)) during postnatal days 21 to 50 alleviated most ASD-like symptoms, including spontaneous alteration, increased locomotion, increased anxiety, and improved social interaction. Treatment also reduced brain oxidative stress, nitrosative stress, and inflammation.

Kim et al. [49] found that the injection of agmatine (an endogenous *NMDA* receptor antagonist) half an hour before the performance of behavioral studies on the rat offspring of dams treated with VPA, and thus demonstrating ASD-like behavior, alleviated their abnormal social behavior, repetitive behaviors, and hyperactivity. Rats were injected on day 12 of gestation with 400 mg/kg body weight of VPA, and the offspring received 25–100 mg/kg of agmatine half an hour before the behavioral studies, which were carried out during postnatal days 26 to 53. Hence, the administration of agmatine in these studies alleviated the ASD-like symptoms and normalized the extracellular signal-regulated kinases *ERK1/2* signaling in the prefrontal cortex and cerebellum.

In a different study, Kim et al. [50] treated pregnant mice and rats with VPA (rats with 400 mg/kg on day 12 of gestation, and mice with 300 mg/kg on day 10 of gestation), and found that the offspring demonstrated autistic-like behaviors. In addition, they observed dysregulation of the cholinergic neurons in the cerebral cortex in the offspring. Therefore, they treated the offspring half an hour before testing with daily intraperitoneal injections of 0.3 mg/kg of donepezil, which is an inhibitor of acetyl choline esterase, in order to normalize the cholinergic system. This treatment of low doses of donepezil improved the sociability of the tested animals, and prevented repetitive behavior and hyperactivity.

#### 4.2.4. Amelioration of Symptoms by Folic Acid in Human Prenatally Exposed to Antiepileptic Drugs

In a very recent study, Bjork et al. [55] explored whether high folic acid supplementation and high blood folate levels in pregnancy are associated with a reduced risk of autistic traits in the offspring of epileptic women receiving antiepileptic drugs. In a Norwegian nationwide cohort study of over 100,000 women, the risk of autistic traits at 36 months of age was significantly higher among children of epileptic women who received antiepileptic drugs without folate supplementation, compared to those who received folic acid supplementation; these findings were inversely correlated with the amount of folic acid ingested and the folate blood levels. There were 38 women exposed to VPA, and four children (12.9%) who had ASD. The increased rate of ASD was not only among those exposed prenatally to VPA, but was also found among the offspring of mothers who received other antiepileptic drugs.

#### 4.2.5. Amelioration of Symptoms by Stimulants

Hara et al. [51] demonstrated that mice prenatally treated on day 12.5 of gestation with 500 mg/kg of VPA exhibited ASD-like behavior that was apparently related to a reduced function of the dopaminergic activity in the prefrontal cortex. These changes were observed in male offspring, but were not observed in females. In a later study [56], they found that the daily administration of stimulants (methylphenidate and atomoxetine) for two weeks increased prefrontal dopamine and norepinephrine release in the prefrontal cortex, and improved the social interaction deficits in these animals. Male behavioral improvement lasted as long as treatment continued.

#### 4.2.6. Administration of Oxytocin to Mice with VPA-Induced ASD-Like Behavior

Dai et al. [52] studied the levels of oxytocin in the hypothalamus of adolescent rats prenatally exposed to VPA and exhibiting ASD-like behavior. They found reduced oxytocin-secreting cells (oxytocin-ir cells) in the paraventricular and supraoptic nuclei of the hypothalamus, as well as reduced oxytocin mRNA. Daily subcutaneous injections of oxytocin to the offspring from birth to adolescence improved their social preference and decreased the repetitive stereotyped behavior. A single intranasal administration of oxytocin half an hour prior to behavioral testing at adolescence restored the impaired social preference. Similarly, Hara et al. [57] administered intranasal oxytocin to mice prenatally exposed to VPA and observed complete restoration of the social behavioral deficits that lasted for about two hours, but had no effect on recognition memory impairment. Oxytocin also increased c-fox expression in the prefrontal and somatosensory cortex, but not in the hippocampus.

#### 4.2.7. Reversal of VPA-Induced ASD-like Behavior in Mice by Human Adipose-Derived Stem Cells and by *S*-Adenosine Methionine (SAM)

These studies seem to demonstrate the real reversal of the ASD-like symptoms and also the reversal of some of the typical pathological markers of ASD. Therefore, they will be discussed at some length [21,53,54].

##### Reversal by Human Adipose-Derived Stem Cells (hASCs)

Ha et al. [53] used human adipose-derived stem cells (hASCs), which are mesenchymal pluripotent stem cells that are considered to have potential therapeutic effects on ASD. They injected *BALB/c* mice with VPA on gestational day 13, and injected hASCs into the brain ventricles of male offspring on postnatal days two and three. The animals were tested by different behavioral tests three weeks after the injection of the hASCs. The researchers observed a significant reduction in repetitive behaviors, improved socialization, and decreased anxiety compared to the animals that were treated prenatally with VPA and not injected the hASCs. In addition, they observed a correction of several biochemical parameters that were deranged in the brain by the administration of VPA, including restoration of the alterations in phosphatase and tensin homolog (*PTEN*) expression, and normalization of the levels of Vascular endothelial growth factor *(VEGF)* and *interleukin-10*.

##### Reversal of Symptoms by *S*-Adenosine Methionine (SAM)

Ornoy et al. [21] injected four-day-old mice with a single dose of 300 mg/kg of VPA, producing ASD-like behavior. The animals were tested on a variety of neurobehavioral tests during postnatal days 50 to 59, and exhibited typical ASD-like neurobehavioral changes. In addition, various parameters of oxidative stress were evaluated in the prefrontal cortex and liver on day 60. Enhanced oxidative stress was observed in the prefrontal cortex, which was manifested by the increased activity of superoxide dismutase (SOD) and Catalase (CAT) enzymes, increased lipid peroxidation, and changes in SOD and CAT gene expression. No changes were observed in the liver, implying that the oxidative stress was induced only in the brain as a result of the epigenetic changes induced by the VPA. In addition, there were also significant differences between males and females in the autistic-like behaviors, as some of these behaviors were more prominent in males (i.e., lower preference for social novelty), while others were more prominent in females (i.e., greater anxiety). In general, brain oxidative stress was higher in females compared to males [21]. VPA also induced changes in gene expression, upregulating or downregulating 29 out of the 770 genes involved in nervous system function and in neuropathology, as evident by NanoString nCounter analysis [54].

VPA-injected pups were also given daily oral treatments of 30 mg/kg of *S*-adenosyl methionine (SAM), a potent physiological methyl donor, for three days from postnatal day five. SAM prevented the neurobehavioral autistic-like changes in both male and female mice, decreased the degree of oxidative stress, and reversed the gene expression changes induced by VPA. Hence, SAM administered closely after the administration of VPA prevented the ASD-like changes in our mouse model. We presume that VPA induced the epigenetic changes in the brain that caused the ASD-like behavior. DNA methylation studies are now being carried out to elucidate the exact mechanism and the possible use of SAM for the prevention of ASD in other animal models.

## 5. Summary of VPA Studies

VPA is a potent teratogen that may increase the rate of major congenital anomalies and neurobehavioral disorders, including ASD. High doses of folic acid in rodents prevented some of the congenital malformations, especially the neural tube defects. However, there is little data on the prevention of VPA-induced neurodevelopmental problems in rodents. The successful prevention of ASD-like behavior in mice and rats was reported following the use of antioxidants, namely astaxantine, piperine, and green tea extracts. Similar results have been obtained by the treatment agents that affect brain neurotransmission such as donepezil, agomelatine, memantine, agmatine, and oxytocin. We prevented the development of ASD-like behavior in mice postnatally injected with VPA by the administration of SAM, which is a physiological methyl donor and antioxidant. Most of the studies that used the above-mentioned agents indicated the amelioration and/or alleviation of the ASD-like symptoms after they appeared. However, these agents were serving more as a type of “medication”, since they had to be given continuously in order to be effective. So far, there seems to be no data regarding successful attempts to prevent VPA-induced neurobehavioral damage in man.

## 6. Animal Models of ASD: Diagnostic Tests

Since ASD is primarily a neurodevelopmental disorder without any specific diagnostic laboratory test, it is imperative to develop tools for the developmental assessment of ASD-like behavior. Once these tools are established, ASD-like models can be developed. Indeed, with the development of neurobehavioral teratology among animals, specific behavioral tests for rodents were developed that are able to measure the different behavioral modalities that are typical of human ASD. The following behavioral tests are often used for the evaluation of autistic-like behaviors, especially in mice and rats. These tests serve as proof of ASD-like behavior. They are aimed at measuring social behavior, cognitive impairment, anxiety, and motor behavior, including repetitive movements. It is important to note that animals with autistic-like behavior do not necessarily exhibit all of the neurobehavioral features observed in human with ASD.

### 6.1. Tests for Measuring Social Behavior and Anxiety

#### 6.1.1. Open-Field Test

The open-field test is commonly used to assess general locomotor activity, exploring behavior, and anxiety-related behavior. Total distance traveled and the time spend in the center of the field are measured to assess the activity of the animals. In addition, the total duration of self-grooming and rearing activity is measured for behavior [58,59]. The open-field test is also used in rats for locomotor activity and for anxiety-like behavior measurements [60,61].

#### 6.1.2. Three-Chamber Social Interaction/Crawley’s Tests

These tests assess sociability and preference for social novelty, and identify rodents with deficits in sociability and/or social novelty. Social preference is evaluated as a ratio between the time spent with a novel social stimulus versus a familiar social stimulus [58,59,62,63].

#### 6.1.3. Elevated Plus-Maze/Elevated Zero-Maze

These two maze tests are used to assess anxiety-like behavior in rodents. Animals are given access to all the arms, and are allowed to move freely between them. The number of entries into the open arms and the time spent in the open compared to the closed arms are used as indices of open space-induced anxiety [59,64].

#### 6.1.4. Reciprocal Social Interaction

ASD-related behavioral deficits are reflected by a reduction of exploratory behavior toward a previously unrecognized, normal animal, matched by strain age and sex. The test is based on measures of free interaction between the exposed and the control animal [59,65].

### 6.2. Tests for Cognitive Impairment (Memory)

#### 6.2.1. T-Maze Test

The T-maze is used to assess the memory and cognitive ability of rodents. This elevated or enclosed apparatus is placed horizontally. Animals are placed on the base of the T, and allowed to choose one of the goal arms. Generally, in the second trial, the animal will chose the arm that was not visited before, reflecting the memory of the first choice. This is called “spontaneous alternation”, and it measures mainly hippocampal function [66]. 

#### 6.2.2. Morris Water Maze

In the Morris water maze, information technology (IT) is used to assess cognitive function by testing the spatial or place learning and memory and reversal memory of animals. Rodents use spatial cues to learn the location of a submerged underwater platform for escape from a circular pool. Once the learning criterion is met, the location of the platform is switched to another location in the pool, and the animal is faced with the task of learning the new spatial location [62,67,68].

#### 6.2.3. Water T-Maze

The water T-maze is used to assess reversal learning, cognitive rigidity, and repetitive behavior. Rodents learn to select a particular arm of the T as to obtain a positive platform. Then, the platform is moved to the opposite arm of the T, and the animal must inhibit the initial learned response and learn the new location of the platform [58,59,69].

#### 6.2.4. Three Arm Y-Maze

This maze with three identical arms is used to assess short-term memory, which is measured by arm and object discrimination. It mainly measures hippocampal functions in mice and rats [70].

#### 6.2.5. Novel Object Recognition

Novel object recognition evaluates cognition, particularly recognition memory, and is based on the rodent’s spontaneous tendency to spend more time exploring a novel object than a familiar one. The time spent exploring novel and familiar objects and the discrimination index percentages are recorded [71,72].

### 6.3. Tests for Locomotor Activity and Repetitive Behavior

#### 6.3.1. Open-Field Motor Activity

Open-field behavior is used for the assessment of alterations in locomotor performance levels. The test focuses on a few measurements that provide an assessment of motor performance. Mice locomotor activity on the open field is recorded using a video camera and the data analyzed for multiple behavioral and motor analysis, including horizontal activity, vertical activity, movement time, rest time, and total distance traveled. In general, mice and rats with reduced muscle function will be less active and have lower ambulatory activity [59,73,74].

#### 6.3.2. Rotarod Performance Test

This test evaluates motor coordination and motor learning in mice and rats, and is especially sensitive in detecting cerebellar dysfunction [75]. Animals are placed on a horizontal rod that rotates around its long axis with forced motor activity. The rodent has to keep its balance and walk forwards on the rotating rod in order to remain upright and not fall off. Motor coordination, balance, grip strength, and motor learning can be tested by comparing the latency to fall on the very first trial with that of subsequent trials.

#### 6.3.3. Marble Burying 

This test is used for the assessment of repetitive behavior. Black marble stones are placed on fresh bedding in the cage. Digging in the bedding and burying the marble stones is recorded for about 15 minutes. The time until over 75% of the marbles are buried is recorded [76].

## 7. Did animal studies improve our knowledge to treat ASD in man?

### 7.1. Translation of Animal Studies to Human ASD

Although many animal studies proved the possibility of partially reversing or temporary ameliorating the ASD-like symptoms, the translation of animal data to human application is met with difficulties. This may be explained by the etiology and pathogenesis of ASD still being poorly understood. Hence, clinical trials based on animal data have been rarely performed. However, patients with ASD often receive various psychiatric drugs for the alleviation of disturbing behavioral deviations. For example, they are treated with antipsychotic drugs or antidepressants if they have psychotic or depressive behavior, or with stimulants if they demonstrate ADHD-like symptoms. In addition, animal data and the understanding of the importance of PUFAs, especially omega-3 and omega-6 PUFAs in brain development [77], prompted the initiation of many clinical studies using PUFAs for the possible alleviation of ASD-like symptoms. Hence, we will describe some of these studies below.

### 7.2. Possible Amelioration in Human by PUFAs

PUFAs are used for treatment trials in several psychopathologies, especially in the treatment of ADHD [77,78]. This apparently stems from a partial deficiency of omega-3 PUFAs being demonstrated in children and adults with ADHD and several other mental disorders, including ASD. Indeed, many clinical trials treating children with ADHD have shown partial success in the alleviation of ADHD symptoms.

There are several clinical trials featuring children with ASD, especially those that have hyperactivity and other symptoms of ADHD. For example, Youi et al. [79] treated seven ASD children with large doses of arachidonic acid and docosahexaenoic acid for 16 weeks and compared them to six children who received placebo. They observed some improvement of social behavior in the PUFA-treated group compared to the placebo controls. Bent et al. [80] treated ASD children with hyperactivity for six weeks with daily doses of 1.3 g of omega-3 fatty acids, compared to placebo. Some improvement in the hyperactivity scores of the omega-3-treated children was observed, but the difference was not statistically significant. Voigt et al. [81] treated children with ASD who were between three and 10 years old with 200 mg/day of docosahexaenoic acid for half a year, but did not find any improvement in autism symptoms.

It is interesting to note that in 2017, two meta-analyses were published with contradictory results. In the one published by Horvath et al. [82] on 183 ASD children supplemented with omega-3 fatty acids, no beneficial effects of the supplementation could be demonstrated on the ASD symptoms. The second meta-analysis was published by Cheng et al. [83], which included six randomized controlled studies (out of 34 studies). The authors found that in 99 treated children with ASD compared to 95 ASD children who received placebo, there was a significant improvement in hyperactivity scores, stereotypy, and lethargy, but no improvement in other functions, especially in social and communication functions.

It can be summarized that the benefit of PUFA supplementation in children with ASD is limited. Although PUFAs possibly improve hyperactivity, there seems to be no effect on communication abilities.

## 8. General Conclusions

To date, there has been no effective pharmacological or medical treatment for ASD, and the etiology is poorly understood. Hence, animal models increase our knowledge not only of the etiology and pathogenesis of ASD, but also of new potential methods of effective treatment. In this review, we discussed the different studies describing attempts to ameliorate or reverse the ASD-like behavior in experimental animals where ASD was induced either by genetic manipulations, inflammation/infection, or prenatal or early postnatal administration of valproic acid. Most studies have demonstrated that antioxidants or substances affecting brain neurotransmission are able to modify or reduce the ASD-like behavioral changes, but these effects were generally only temporary. Only a few studies have shown a long-term reversal of the ASD-like behavior. So far, there have only been a few attempts for the translation of the animal data to human ASD. It is possible that the animal data on the long-term reversal of ASD-like symptoms, if explored further, may prove to be beneficial to human ASD, but we still have a very long way to go before appropriate clinical trials are implemented.

## Figures and Tables

**Table 1 ijms-20-01074-t001:** The different studies demonstrating the amelioration or reversal of symptoms in valproic acid (VPA)-induced autism spectrum disorder (ASD)-like behavior. The order is the appearance in the text of the review.

Agent Used	Author	Mode of Treatment	Animal	Outcome
Green tea	Banji et al., 2011 [42]	VPA post natal day (PND) 14 Green tea daily 75 mg/day or 300 mg/day	mice	Improved behavior and prevention of histopathological damage to cerebellum,
Astaxanthin	Al-Amin et al., 2015 [43]	VPA gestation day 12.5; astaxanthin days 25–40	mice	Improved behavior and reduced brain oxidative stress
Piperine	Pragnya et al., 2014 [44]	VPA PND 14; piperine PND 13–40	mice	Improved behavior, reduced oxidative stress, elevated brain glutathione
α or γ linoleic acid	Yadav et al., 2017 [45]	VPA PND 14, α or γ linoleic acid daily	rats	Decrease in markers of oxidative stress and improved behavior
Agomelatine	Kumar et al., 2015 [46]	VPA gestation day 12.5; agomelatine PND 21–50	rats	Improved behavior
Minocycline	Kumar et al., 2016 [47]	VPA gestation day 12.5; minocycline PND 21–50	rats	Improved behavior
Memantine	Kumar et al., 2016 [48]	VPA gestation day 12.5; minocycline PND 21–50	rats	Improved behavior
Agmatine	Kim et al., 2017 [49]		mice	Improved behavior; normalized *ERK1/2* signaling in the prefrontal cortex and hippocampus
Donezepil	Kim et al., 2014 [50]	Prenatal VPA; donepezil given daily, days 14–40	mice and rats	Improved behavior; reduced repetitive movements and hyperactivity
Methylphenidate, Atomoxetine, Oxytocin	Hara et al., 2015 [51]	Prenatal VPA; daily treatment with stimulants	mice, males	Improved behavior; increased prefrontal dopamine, and norepinephrine levels
Oxytocin	Dai et al., 2018 [52]	Prenatal VPA; daily oxytocin injections: birth, adolescence, or single intranasal dose	rats	Improved behavior; after single dose, improvement lasted for two hours
Human adipose derived stem cells	Ha et al., 2017 [53]	VPA day 13 of gestation; brain intraventricular injection of stem cells	mice	Improved behavior: reduced anxiety, better socialization, normalization of some biochemical abnormalities
*S*-adenosine methionine (SAM)	Ornoy et al., 2018 [54]	VPA PND 4, SAM PND days 5–7	mice	Improved behavior, decreased brain oxidative stress, reversal of abnormalities in gene expression induced by VPA

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
