# Peer review of "Prevention or Amelioration of Autism-Like Symptoms in Animal Models: Will it Bring Us Closer to Treating Human ASD?"

_ijms, 2019, doi:10.3390/ijms20051074_

Round 1
Reviewer 1 Report
The comments were properly answered
Reviewer 2 Report
The final version of the manuscript is improved.
This manuscript is a resubmission of an earlier submission. The following is a list of the peer review reports and author responses from that submission.
Round 1
Reviewer 1 Report
This review is describing the agents that ameliorate the autistic behaviors in the animal models in the VPA-induced and genetically modified mice. However, the authors are not carefully reading the references and many statements are wrong or not appropriately reflecting the references. Also, there are many careless mistakes for citing the references (ex. numbers of the ref). It is not my job to check through each references if it is not correctly cited or not. I will point out some mistakes I found as examples, but authors must go through the references themselves again, because I assume there should be more. The authors must carefully reexamine what they need to state from the references. This point is very fatal for such review article.
Examles:
1.Page2 line 60 and 61
Authors misreading Ref 6. It is checking grooming behavior in the wheel test and not in the open field test. Furthermore, it is not common to measure the self-grooming in the open field test. May remove the self-grooming. Segal- Gavish et al, is not published in 2015.
2. Page 4 line 140
I assume authors wanted to cite Zhang et al (ref 24) and not Teng et al. Furthermore, authors mentioned the function of oxytocin as mother-infant bonding. However, the reference 24 is not investigating the mother infant bonding at all. The statement is misleading as if the ref is investigating the mother infant bonding.
3. Page 4 line 167
Authors state that Avraham et al treated newborns with beta carotene to the newborns that exhibit autistic behavior. Avraham et al treated beta carotene to newborn babies without testing abnormal behaviors.
Reviewer 2 Report
In the present work Ormoy and colleagues aimed at evaluating the role of animal models in the understanding of preventive strategy for ASD in humans.
The work is interesting, but it needs to be reformulated substantially.
the paragraph n°2 is not justified in the order of appearance I suggest to move it in further sections. In relation to this paragraph I do have some concerns:
- in the tests for cognitive impairment: no mention is made of other very important e widely used tests such as T maze or Y maze
- for open field test: such test does not measure muscle function or muscle strengh, but it can only allow to identify a possible alteration in locomotion that can be caused also by other factors drug-induced such as catalepsy or sedation. furthermore, this test is widely validated also in rats, thus it cannot be referred only to mice.
in the part 3.1, notions about genetic animal models should be expanded extensively. BTBR mice models has been studied in many behavioral paradigms that deserve to be mentioned in a review (for example https://doi.org/10.1016/j.bbr.2018.02.003 and so on)
3. in paragraph 4.2 please add the study about PUFA and ASD, (you can refer for your convenience to doi: 10.3390/brainsci6030024; doi: 10.1001/jamapsychiatry.2014.476.)
4. the paragraph 5.1 is referred to humans, however only half paragraph is related to humans while the other half part is again focused on animals. Thus, I suggest either to change the title or, preferably, to expand on human studies.
5. In the paragraphs related to ameliorative treatments, please be more precise by adding treatment schedules and dosages.